# Changing Social Mentality among University Students in the COVID-19 Pandemic: A Five-Wave Longitudinal Study in China

**DOI:** 10.3390/ijerph19053049

**Published:** 2022-03-05

**Authors:** Jingjing Zhao, Mengyu Yan, Bingbing Fan, Yueyang Zhang, Anwar Oguz, Yuying Wang, Juzhe Xi

**Affiliations:** 1School of Public Health, Cheeloo College of Medicine, Shandong University, Jinan 250012, China; zhaojingjing@sdu.edu.cn (J.Z.); yanmengyu2001@163.com (M.Y.); zyy15809057928@outlook.com (Y.Z.); agzsdu811@163.com (A.O.); yingkamn@163.com (Y.W.); 2School of Marxism, Shandong University, Jinan 250100, China; 3Department of Biostatistics, School of Public Health, Cheeloo College of Medicine, Shandong University, Jinan 250012, China; fanbingbing92@163.com; 4Harvard College Association for US-China Relations (HCAUSCR), Harvard University, Cambridge, MA 021382, USA; 5Shanghai Key Laboratory of Mental Health and Psychological Crisis, Affiliated Mental Health Center (ECNU), Positive Education China Academy, Juzhe Xi’s Master Workroom of Shanghai School Mental Health Service, School of Psychology and Cognitive Science, East China Normal University, Shanghai 200062, China

**Keywords:** pandemic, university students, social mentality, mental health, longitudinal study

## Abstract

(1) Background: The COVID-19 pandemic has led to significant shifts in university students’ lives, which could be displayed by social mentality, a psychosocial conception at the individual and social levels. This five-wave longitudinal study aims to evaluate the changing social mentality of university students during the peak and preventive-order phases of the pandemic in China and investigate the trends and differences in social-demographic variables. (2) Methods: The Bi-Dimensional Structure Questionnaire of Social Mentality (B-DSMQ) was used to collect data from March 2020 to January 2021. Five-wave surveys were administered to 1319 students from five universities using online questionnaires. Analysis of variance (ANOVA) was used to compare the changes in social mentality over time and covariate groups. Linear mixed models were used to explore the associations of overall social mentality with time and covariates. Post hoc analysis was implemented within subgroups, including university, major, grade, parenting style, and the harmonious degree of parents. (3) Results: Students’ social mentality changed significantly from Waves 1 to 5 (*p* < 0.001). It fell to its lowest in the third survey, increased in the fourth survey, and peaked in the fifth survey. In all of the subgroups, the changing social mentality differed significantly over time (*p* < 0.001). The *p*-values between groups suggested that changing social mentality was significantly different regarding gender, residence, university, major, grade, student cadre, graduates, nuclear family, economic status, parenting styles, and the harmonious degree of parents’ relationship (*p* < 0.001). (4) Conclusions: Social mentality among university students decreased during the peak of the pandemic before increasing in the contained-risk period. It was the lowest in June when students began to return to the pandemic-preventive campus from quarantined homes. Students living in provinces (except for Shandong) who were from high-level universities in 2016 and 2017 and who majored in medicine displayed a more negative social mentality. Students who were female, student cadres, non-graduates, and enjoying high socioeconomic status displayed a more positive social mentality. Further research is needed on the relationship between mental health and social mentality, specifically the associates and interventions for positive social mentality.

## 1. Introduction

Since the discovery of the novel coronavirus disease (COVID-19) in December of 2019 and the World Health Organization officially declaring it a pandemic on 11 March 2020 [1], the virus continues to pose a substantial global risk to public health and disrupts lives on an unprecedented scale. For many universities worldwide, lockdowns have led to significant shifts in students’ lives. Available research has proven that university students experience an atmosphere of cognitive distress and negative emotions due to social isolation, media information overload, reduced sleep quality, and a sense of numbness [2,3]. This environment, according to psychosocial research, is called social mentality. As a psychosocial concept, social mentality is not the simple accumulation of individuals’ mental health status but is constructed by them through deindividuation and social identification [4,5]. As a barometer of social stability, the mentality has had a substantial influence on individual mental health and social mobilization at the individual and group levels during the COVID-19 pandemic.

### 1.1. Conception of Social Mentality in This Study

The study of social mentality focuses primarily on history, philosophy, and social psychology. The term “mentality” was coined in historiography. There are two primary definitions in Larus’s French Dictionary: one is the spiritual customs, beliefs, and emotions of a group, and the other is the individual mental state (or psychological state) [6]. It originated in British philosophy and refers to the psychological characteristics shared by a nation or a human group, expressed in sensory and mental ways. Social psychology is a common concept in the tradition of historical materialism, and social mentality and social psychology are inextricably linked. In the 1990s in China, some scholars debated the issue of social mentality, beginning with historical materialism [7]. They were convinced that “social mentality research aims to elucidate the spiritual intermediary between social subjects and objects. There are two levels of mediation between social subjects and objects: material mediation (practice) and spiritual mediation (social mentality).”

Research on social mentality in social psychology began in two major areas. In North America, it is primarily concerned with the process by which individuals are passively influenced by others (i.e., how groups actively influence individuals) [8]. Additionally, European social psychology examines individuals’ active integration into groups [9]. Yang Yiyin (2006) from China combined the two approaches and proposed a ring structure from individual to group, namely, the interactive construction process model, including individual psychology and social mentality [10]. The model is a dynamic, active system wherein social mentality is an influence variable and a process and outcome variable. The model aims to reveal the mutual construction of the individual and society in most macro psychosocial relations. According to social psychology research, social mentality refers to the cognition, emotions, values, and behaviors permeating the entire society or social group or category over a period [10]. Specifically, it displays the general state of mind at the group level, interactively constructed by individuals and society, with the characteristics of changeability, universality, simplicity, emotionality, and infectiousness [11]. It is not a simple combination of numerous individual-specific responses but constructed among individuals by social identification to exhibit a similar mood at the group level.

Questionnaires are widely used to assess social mentality in multilevel dimensions, and descriptive analysis is frequently applied to statistical analysis [12]. However, due to diversity in individual reactions and the heterogeneity of samples, it is difficult to obtain comprehensive knowledge of social mentality. The social mentality should be investigated at macro and micro levels. As a new trend, resilience and positive psychology seek to discover and stimulate healthy life vitality as a personal responsibility to investigate how people actively participate successfully in social change and maintain a healthy development function, thereby providing research on social mentality with a promising paradigm for reference. Furthermore, it is important to examine positive adjustment among young adults during public health emergencies such as COVID-19. Therefore, the conceptual framework design of this study incorporates two considerations and breakthroughs.

First, while considering the macro–micro level, we also consider the “positive–negative” mentality, where one dimension is “individual–public” and the other is “positive–negative”. The former is the subject dimension of social mentality, encompassing micro and macro; the latter is the valence dimension, comprising negative and positive. 

Second, similar to how positive psychology introduces the emotional balance index when examining the emotional dimension of happiness [13], we believe that the social mentality balance index should be introduced when examining the characteristics of social mentality. The emotional balance index was calculated by subtracting the difference between positive and negative emotion, that is, BA = PA − NA, where BA represents the emotional balance index score, PA represents positive emotion, and NA represents negative emotion. The emotional balance index measures the balance between positive and negative emotions. Similarly, we can define the social mentality balance index, as expressed by the formula: BSM = PSM − NSM, where BSM denotes the social mentality balance index, PSM denotes the positive social mentality score, and NSM denotes the negative social mentality score. The addition of a social mentality balance index would enhance the investigation. Simultaneously, it effectively accounts for two distinct social mentalities and integrates them.

### 1.2. Impact of COVID-19 on Students’ Social Mentality

As the pandemic continues to evolve, young adults are particularly vulnerable [14]. Extra-curricular activities were postponed or canceled, and face-to-face courses and internships were not possible [15], leaving students with an aggravated psychosocial status [16]. Researchers and policymakers have increasingly focused on the mental health status and adjustment caused by school closures and home confinement. In a previous study by this research team, we demonstrated the short-term psychological impact of the pandemic on the home-quarantined university students, revealing the presence of post-traumatic stress [17]. A longitudinal study by Wu S.Z. et al. focused on changes in students’ mental health in response to the pandemic and local factors’ role in these changes [18]. The results from the research above provided strong evidence for the knowledge of students’ mental health and the implementation of psychological interventions during the pandemic. However, due to its impact on the pandemic on politics, economy, culture, society, and lifestyles, it is necessary and worthwhile to make efforts regarding the knowledge of mentality specific to students, including values, attitudes, cognition, and emotions, from a macro perspective.

Although mental health and social mentality are evaluated using individual indicators, the latter evaluates the psychological state of the individual, along with that of the group from the group members’ perspective, which reflects the construction of the individuals’ and the public’s perspectives. According to Yang’s (2006) interaction model of social mentality (introduced in Section 1.1), individual psychology and social mentality interact in the COVID-19 context. Anxiety, depression, and stress are influenced by interpersonal interactions and group identification. In other words, an individual’s state of mind is influenced by group values and attitudes. Therefore, based on research on mental health, it is necessary to investigate the psychological state at the group level to improve group intervention and guidance, create a positive social atmosphere, and promote social stability and harmony.

Most research on university students’ social mentality during the pandemic was cross-sectional [19]. Four features were highlighted among this group: worrying about their health, sadness and anger when watching the news, burnout, anxiety due to remote learning at home, as well as anxiety and loss due to employment and academic pressure [20]. Regarding the influencing factors, students’ social mentality was not only influenced by families, universities, and the social environment at a macro level, but also reflected individual cognition, emotions, values (e.g., the sense of meaning in life), and behaviors at the micro level [21,22,23]. In other words, the influencing factors are complex and multiple, including environmental and individual psychological factors [24].

Although we are starting to understand the immediate effects of the pandemic on university students’ social mentality, little is known about how it evolves. Changes in social mentality reflect the pandemic’s significant effect on society and resilience among individuals at the group level [25,26]. Available longitudinal research was implemented among the general population within one month of the outbreak [12,27]. Changes in social mentality within one week after the outbreak of COVID-19 showed that risk perception tended to be rational, emotions (both positive and negative) fluctuated, and more and more people took protective actions as well [12]. Continuous negative emotions have stressed the public for many years. Furthermore, changes in social mentality in the epidemic area were different from those in other areas [27].

To the best of our knowledge, no longitudinal study has explored the dynamics of social mentality containing subject (individual–public) and valence (positive–negative) across the different stages of the pandemic. Furthermore, none have investigated the same individuals at more than four time points during the pandemic. Only one four-wave study by Li et al. explored emotional changes across various stages of COVID-19 in the United States and China. The emotional recall task (ERT) was used to investigate individuals’ emotions and the valence (positive–negative) of emotion. Moreover, they measured life satisfaction, preventive behaviors, the acquisition of COVID-19 related information, and risk perceptions [28]. Their research suggests a possible life cycle of emotional reactions among individuals toward a pandemic and highlights the importance of people acquiring information and knowledge about the threat in containing its spread. As emotionality is a characteristic of social mentality, familiar emotions among a specific group are constructed through interpersonal communication and identification. It would be helpful to know the entire picture of individuals and the public, which “social” specifically refers to. This study attempts to examine the changing social mentality specific to university students during COVID-19 to complement the aforementioned research. 

This five-wave longitudinal study aims to evaluate the changing social mentality of college students during the peak and preventive-order phases of COVID-19 in China, investigate the trends and differences in university students’ social mentality in social demographic variables, and provide further empirical data on university students’ social mentality. Along with the spread of the risk, the possibility of going back to the university campus was a critical environmental situation in this study. The results could be used as a reference in the provision of effective mental health education and interventions for universities during public health emergencies.

## 2. Materials and Methods

### 2.1. Participants

Students of 30 majors from 5 universities in Shandong province were recruited through stratified cluster random sampling. Firstly, we randomly selected five universities from Shandong Province, including one top comprehensive university and four ordinary universities. Secondly, we randomly selected six majors from all majors in each university. Thirdly, one class from each grade of each major was randomly selected, respectively. We conducted a baseline online survey and 4 follow-up online surveys for 10 months. Data in each wave were summarized in Table 1, including wave/time/number of responses obtained/recovery rate. A total of 1319 university students aged 16–28 years were finally included in this study.

### 2.2. Study Design and Data Collection Procedure

A five-wave survey was designed to track university students’ social mentality as the COVID-19 pandemic unfolded. As is shown in Table 1, the first wave of data collection took place on 3–10 March 2020, when China was experiencing the full force of the COVID-19 pandemic and all of the university students were home-quarantined [29]. The second wave took place on 8–15 April 2020, after home-quarantined students had attended online classes for about two months. The third wave took place on 17–24 June 2020, when graduating students returned to campus [30]. The fourth wave took place on 1–6 November 2020, when back-to-campus non-graduating students had studied in pandemic-preventive order for about two months. It was worth mentioning that from April (Wave 2) to November (Wave 4), the risk of the epidemic was mainly contained in China but ongoing all over the world [31]. The fifth wave took place on 18–25 January 2021, when China’s government started to provide COVID-19 vaccines for all citizens while students went home after completing the autumn-term study on campus [32].

Considering the limited accessibility to respondents owing to the social distancing policy in effect during the COVID-19 pandemic, all data were collected using “Wenjuanxing”, an online survey platform in mainland China. Full-time tutors who majored in psychology, education, and administration were trained online on the aim of the study, data collection procedures, and privacy protection policies during the survey at the beginning of the survey. A link to the questionnaire was sent to potential respondents in sampled classes by tutors through WeChat, a popular mobile app in China. Each IP address was allowed one questionnaire response. Statements of the purpose of the research and assurance of the confidentiality and privacy of participating individuals were placed on the first page of the survey questionnaire. After reading this statement, participants could only complete the questionnaire by clicking “AGREE” to confirm their consent. All participants were told that they had the right to stop the survey at any time. Respondents could get lucky money after finishing the online survey in each wave. Those who filled in the first-wave questionnaire were followed-up within the successive four waves by Wechat. One trap question (e.g., Select “Satisfied” in this question to check the level of careful reading) in each survey was set up to check if the respondent was reading carefully. Responses choosing the wrong answer were considered illogical. After excluding illogical or missing responses, 1319 respondents who completed all five surveys formed the final database.

### 2.3. Measurements

#### 2.3.1. Social Demographic Variables

The demographic information included individual, school, and household variables. Individual-level variables included age, gender, ethnicity, student cadre (Yes/No), and graduating student (Yes/No). School-level variables included university, grade, and major. Household-level variables incorporated family styles, social–economic status, parenting styles, residence, and parents’ relationship.

#### 2.3.2. The Bi-Dimensional Structure Questionnaire of Social Mentality (B-DSMQ)

We used the Bi-Dimensional Structure Questionnaire of Social Mentality (B-DSMQ) to measure social mentality. The scale is designed in two dimensions, where one dimension is “individual–public” and the other is “positive–negative.” The former is the subject dimension of social mentality, encompassing both micro and macro; the latter is the valence dimension of social mentality, encompassing both negative and positive. The scale contains 46 words or phrases (21 for positive social mentality and 25 for negative social mentality). Meanwhile, 25 of those phrases above were used for the individual mentality and 34 for the public mentality (see Appendix A). The B-DSMQ includes positive individual mentality (PIM), negative individual mentality (NIM), positive public mentality (PPbM), negative public mentality (NPbM), positive social mentality (PSM), negative social mentality (NSM), and balanced social mentality (BSM). PSM is comprised of PIM and PPbM, while NSM is comprised of NIM and NPbM. BSM, which is the balance between PSM and NSM, that is, BSM = PSM − NSM, indicates the proportion of positive social mentality and describes the overall social mentality based on the subject (individual–public) and valence (positive–negative). The scale applied the item presentation method in PANAS (Positive and Negative Affect Schedule, a widely used scale evaluating emotions), displaying specific phrases of social mentality, such as being insecure, being hopeful, being faithful, being tolerant, being harmonious, being unfair, being honest, being grateful, and being supportive. The subjects were asked to rate these items from their standpoint and the public’s point of view. Responses were rated on a 6-point Likert scale, ranging from 1 (not at all) to 6 (mostly). Higher BSM indicates that the overall social mentality is more positive. The validity and reliability have been confirmed in previous studies [11]. The Cronbach’s alpha of B-DSMQ in this study ranged from 0.917 to 0.970.

### 2.4. Ethical Statement

The project was approved by the Institutional Review Board of the School of Public Health, Shandong University. The corresponding ethical approval code was LL20200201. Informed consent was obtained from all respondents.

### 2.5. Statistical Analysis 

Analysis of variance (ANOVA) was performed to test if the social mentality varied with time. The null hypotheses of the tests were that the seven social mentalities did not change with time (wave). The *p*-values of the hypothesis tests were then adjusted by Bonferroni-Holm correction (α= 0.007). The change of balanced social mentality (BSM) over group variables was tested by repeated-measures analysis of variance (RM-ANOVA), with one group variable and time (wave) tested in one model. The null hypotheses of the RM-ANOVA were: (1) the BSM did not change over time; (2) the mean values of BSM did not differ over groups. Additionally, we adjusted the *p*-values by Bonferroni-Holm correction (α = 0.004) (Table 1) for Type 1 error protection. Post hoc analysis was used in order to compare the difference of social mentality among five waves and explore specific changing trajectories from Waves 1 to 5. The same method was used to display the difference within subgroups, including university, major, grade, parenting style, and harmonious degree of parents. Linear mixed models were used to explore the associations of overall social mentality with time and covariates. All analyses were implemented with R software (version 4.0.1) (R Foundation for Statistical Computing, Vienna, Austria).

## 3. Results

### 3.1. Sample Characteristics

The analytical sample for our longitudinal analyses included 1319 participants (66.1% females) from 5 different universities who had participated in all 5 waves. Thirty majors were classified into six disciplinary categories, including Engineering, Science, Agriculture, Literature, Art, and Medicine. The average age was 20.1, and 77.25% of the students lived in Shandong Province. Displays of sample characteristics in social mentality based on different levels of BSM in the first wave can be found in Table 2.

### 3.2. The Changes of University Students’ Social Mentality from Waves 1 to 5

Considering that social mentality was based on subject (individual–public) and valence (positive–negative), seven indicators were examined (see Table 3). Among them, the average PIM, NPbM, NSM, and BSM values changed significantly from Wave 1 to Wave 5 (*p* < 0.001). The average BSM value fell to the lowest in the third survey, increased in the fourth survey, and peaked in the fifth survey. The average values of NPbM and NSM rose to the highest level in the second survey and dropped from the third survey to the lowest level in the fifth survey. The average PIM value increased in the second survey, dropped to the lowest level in the third survey, and rose to the highest level in the fifth survey.

BSM, the dependent variable in statistical analysis, as a result of the balance between positive social mentality (sum of individual positive mentality and public positive mentality) and negative social mentality (sum of individual negative mentality and public negative mentality), was considered as the indicator of overall social mentality. Paired *t*-tests in post hoc analysis were used to examine the difference of BSM between waves in detail (See Table 4). Compared to Wave 1, the social mentality of Wave 2 and Wave 3 decreased significantly (*p* < 0.001). Compared to Wave 2, the social mentality of Wave 3 decreased significantly (*p* = 0.001). Compared to Wave 3, the social mentality of Wave 4 increased significantly (*p* < 0.001). However, the increase from Wave 1 to Wave 5 was not significant.

### 3.3. Changes and Differences of Social Mentality in Socio-Demographic Variables in the Sample

In all of the subgroups, the changing BSM of students differed significantly over time (*p* < 0.001 for all). To be more specific, a significant decrease in social mentality from Wave 1 (2.63 ± 1.38) to Wave 3 (2.35 ± 1.59) was observed in females, and it rose to the highest point in Wave 5 (2.74 ± 1.64). Additionally, a significant decrease in social mentality from Wave 1 (2.20 ± 1.43) to Wave 3 (1.85 ± 1.68) was found in students living in provinces, except for Shandong, and it rose to the highest point in Wave 5 (2.43 ± 1.75). Moreover, a significant decrease in social mentality from Wave 1 (2.00 ± 1.53) to Wave 3 (1.49 ± 1.41) was observed in high-level universities, and it increased in Wave 4 (1.82 ± 1.50), decreasing in Wave 5 (1.80 ± 1.54). A significant decrease to the lowest point in social mentality from Wave 1 (1.81 ± 1.37) to Wave 2 (1.13 ± 2.04) was observed in medical students, and it rose to the highest point in Wave 5 (2.48 ± 1.56). There was a significant decrease to the lowest point in social mentality from Wave 1 (2.58 ± 1.43) to Wave 3 (2.27 ± 1.67), and a rise to the highest point in Wave 5 (2.72 ± 1.68) was observed in non-graduate students. The point of the nuclear family in Wave 1 was 2.57 ± 1.43, varying in Wave 3 (2.28 ± 1.66), which was the lowest point of all five waves, and it was incredibly increased from Wave 3 to Wave 5 (2.72 ± 1.69). The points of high social–economic status from Wave 1 to Wave 5 were 2.67 ± 1.44, 2.50 ± 1.59, 2.37 ± 1.69, 2.68 ± 1.61, and 2.79 ± 1.70, which showed a trend of falling first and then rising. A significant decrease to the lowest point in social mentality from Wave 1 (1.85 ± 1.51) to Wave 2 (1.39 ± 1.63) was observed in permissive parenting styles, which rose to the highest point in Wave 5 (1.87 ± 1.90). However, the change in the worse degree of parents’ relationship demonstrated a trend of rising from Wave 1 (1.20 ± 1.62) to Wave 5 (1.90 ± 1.70) (see Table 5 and Figure 1, Figure 2, Figure 3, Figure 4 and Figure 5).

The *p*-values between groups suggested that changing social mentality was significantly different in gender, residence, university, major, grade, student cadre, graduates, nuclear family, economic status, parenting styles, and harmonious degree of parents’ relationship (*p* < 0.001), while the difference in ethnicity was not significant. Students living in provinces, except for Shandong, displayed a more negative social mentality. Moreover, those who were female, student cadres, with high economic status, non-graduates, and in nuclear families displayed a more positive social mentality (see Table 5 and Figure 1, Figure 2, Figure 3, Figure 4 and Figure 5). Post hoc analysis was used within subgroups, including university, major, grade, parenting style, and harmonious degree of parents. During Waves 1 to 5, social mentality among students in University A, who majored in medicine, was significantly lower than those in other groups. As for different grades, social mentality among students in grades 2018 and 2019 was significantly higher than those in 2016 and 2017, while there was no difference between those in grades 2016 and 2017. Students whose parents were authoritative had more positive social mentality than those in other groups. Results from the degree of parents’ relationship showed that the more harmonious the relationship the students’ parents had, the more positive the social mentality their children displayed (see Appendix A).

## 4. Discussion

### 4.1. Changing Characteristics of University Students’ Social Mentality at Different Stages

Results of the coexistence of PSM and NSM in this study are consistent with recent studies [19,24]. However, the social mentality among home-quarantined students decreased during Waves 1 and 2, which provides complementary evidence [12]. From 19 February to 20 March 2020, the severity of COVID-19 increased and gradually peaked [32]. China’s Ministry of Education (MOE) announced that the 2020 spring semester for schools would be postponed. The baseline data for this study were collected in the first month of the spring semester (from 3–10 March 2020). University students had been quarantined since the COVID-19 outbreak and had started online courses after the winter holiday when Wuhan was locked down. In this study, students reported negative mentality, comprising tension, worry, anxiety, and helplessness, which is consistent with recent studies [33,34]. The results from March to April 2020 revealed that positive social mentality tended to decline slowly, while negative social mentality showed an evident increase, suggesting that overall social mentality decreased among home-quarantined students. In a two-wave study by C.Y. Wang et al., the general population in China showed no significant longitudinal changes in stress, anxiety, and depression levels during the initial COVID-19 and the peak four weeks later [35]. In another four-wave study by Y. Li et al., emotional changes were investigated among a Chinese sample from 13 to 17 February 2020 and 8 January 2021. They found that affective states improved between Waves 1 (February 13–17) and 2 (April 5–9) and remained stable from Waves 2 to 3 (July 9–13), indicating that people’s mentality became more positive in this stage [28]. Compared with the aforementioned studies, this study found a significant decrease in social mentality among university students in the same period, suggesting that much more attention should be paid to the youth group. On 19 March 2020, no new confirmed cases were reported in Hubei Province for the first time, indicating that COVID-19 was in control. Thus, the majority of young students reported significantly boosted confidence. However, the first confirmed inbound case posed a new challenge for virus control, requiring a shift in focus to prevent inbound cases and domestic resurgence. Furthermore, challenges regarding learning, interpersonal relations, and the future were different from those of the other groups. According to the literature [18,35,36], the reasons for the aforementioned changes in social mentality could be anxiety and insecurity caused by risk perception in disease outbreaks, loneliness, and helplessness elicited by isolation and social distancing, maladaptation to online learning, tremendous employment pressure in a disrupted job market, tension with families during home quarantine, the reopening of schools affecting students with fundamental psychological problems, and constantly receiving negative information about the pandemic.

When some students returned to their campuses, social mentality during Waves 2 to 3 continued to decrease as PSM decreased and NSM increased. As universities across the country resumed school, Shandong Province allowed some graduates and undergraduates to return to school in batches in mid-May. The third survey was conducted from 17 June to 24 June (one month after the students returned). On 11 June, new cases forced Beijing to upgrade its COVID-19 emergency response from Level 3 to 2. Prevention and control situations in universities became relatively tense. According to the follow-up results between April and June, the positive mentalities of individuals and the public dropped dramatically, while the negative mentalities declined only slightly after a remarkable increase in the last survey. It was consistent with cross-sectional research on psychological stress among back-to-school students [37,38]. Universities devoted their energy to mental health education, crisis intervention, and employment guidance, but this was not as effective as expected. Providing psychological counseling through the internet or psychological hotlines impairs the natural effects to some degree. Graduates were more anxious about what they viewed as one of the worst job markets. Non-graduates had mixed opinions on the effectiveness of online courses and had to cope with the stress regarding their final exams. However, researchers also examined protective factors against psychological stress, such as a high confidence level in doctors, perceived survival likelihood and low risk of contracting COVID-19, satisfaction with health information, and personal precautionary measures [36]. Compared with research on the impact of the pandemic, focused on the negative mentality such as depression, anxiety, and stress [39,40], this study showed positive mentalities such as meaningfulness, hopefulness, insistence, honesty, and necessary knowledge on the optimistic facets of university students’ mentality.

From the summer holiday at home to back-to-school learning in the pandemic-preventive order (Waves 3–4), social mentality began to significantly increase from the lowest point, with PSM increasing and NSM decreasing. China implemented regular prevention and control, while the international situation was more serious [31]. Effective control made it possible for all university students to return to school after a six-month stay-at-home life. The changing social mentality from June to November 2020 showed that PSM was notably boosted and NSM vastly reduced, implying a more positive social mentality. This is consistent with a recent study’s results [41]. The reasons could be from the confidence in the contained-risk environment and regular activities on the campus. Combined with online and offline activities, universities’ rigorous prevention and control efforts protected students from being infected and helped re-establish a regular campus life.

Since vaccination was available, social mentality continued to rise, with PSM in the fifth survey being slightly higher than that at baseline, while NSM was slightly lower. In December, China released its COVID-19 vaccines to the international market after the third-stage clinical trials. A month later (from 18 January 2021), the fifth survey showed that the PSM and OSM values were significantly higher than those in the first wave, suggesting that university students had become more mature and positive. This could be because the adversity made individuals grow, specifically, the youth whose values and personalities were constructed by self-seeking and environmental influence. According to researchers, stressors may also be an important opportunity for building resilience, with downstream benefits for well-being and mental health [42]. Therefore, it is worthwhile to further examine the association between positive social mentality and resilience among university students during this pandemic. However, a specific lag effect may exist, which requires further exploration.

### 4.2. Changing Characteristics of University Students’ Social Mentality among Subgroups

This study found that female students and those living in other provinces displayed higher social mentality in the five waves. Some researchers concluded that women indicated more negative mentality and needed more attention [25,42,43,44]. However, other studies showed no significant differences between genders [45]. There was no conflict regarding the findings because the former stressed individual emotions, whereas the latter concentrated on individual cognition, coping styles, and behavior. This study is an essential extension of the aforementioned research from the perspectives of the individual and the public. According to the authors’ work experiences with university students, although female students experienced negative emotions during the pandemic, they spent more time cooking, taking care of siblings, and narrating their stories to friends or roommates. Furthermore, some research found that they were more optimistic [46]. Longitudinal research is needed to determine whether variables such as positive values or behaviors led to a more positive social mentality among females during the pandemic.

It is worthwhile to comment on socioeconomic status and regional differences in this study. The social mentality among students with low socioeconomic status was less positive. University students in Shandong Province presented a higher level of positive social mentality than those in the other provinces. Zhao believed that regional differences could be observed in young people’s social mentality [47]. Yi et al. explored the regional differences during COVID-19 through cluster analysis [48], and the number of confirmed cases, survival rate, and mortality in Shandong Province were the lowest among all the provinces. In this study, regional differences in epidemic situations, prevention, and control affected social mentality. Shandong Province took appropriate action against COVID-19. This might explain why students in this area reported a better mentality. This finding highlights the importance of considering populations from underdeveloped countries, as they might not have adequate economic, social, and educational conditions, which would affect their ability to stop the spread and fight against COVID-19.

This study found that students in B, C, D, and E (ordinary universities) displayed better social mentality than those in A (top university). Students in A were more likely to experience anxiety and worry because of fierce competition. This result confirms previous studies’ conclusions [49]. This could be caused by uncertainty in learning, the environment, and campus activities during the pandemic. Furthermore, tough competition and heightened expectations would decrease their positive social mentality, while medical students displayed even poorer social mentality than those majoring in other disciplines. Several studies have reported similar results [16,37,50]. A deeper understanding of COVID-19 would help improve the public’s sense of security and mood [51]. However, medical students seemed to have higher risk perception and academic pressure and lower career expectations. They were faced with challenges such as sudden changes in their training routine, decreased patient contact and interactions with peers, and increased risk of contracting the infection, mainly among the students in clinical postings. Therefore, it is much more critical for medical students to accept psychological intervention and humanistic education during the pandemic. However, after they returned to campus, their social mentality returned to the baseline level, which showed resilience among this group.

The students in this study were graduates and non-graduates. The results showed that non-graduates rated social mentality better, consistent with previous research [52,53]. Graduates struggled to adapt to the transition from school to work under academic pressure. They also faced many challenges, such as postgraduate entrance examinations and employment. Anxiety and depression attributed to these factors were exacerbated by the outbreak. Therefore, it was inevitable for graduate students to report a descending overall mentality. However, the juniors became graduates in Wave 4, and they did not show a better social mentality than those who graduated before them. Therefore, importance should be given to students in higher grades in the pandemic-preventive order. They need more active employment guidance and career education to improve their social mentality.

Some evidence suggests that young adults returning home are more vulnerable to depression [5,54]. The pandemic caused students to quarantine at home for almost four months in China. Family style, parental relationships, and parenting style were considered family variables. The results suggested that a positive home situation was important for students to maintain a positive mentality. Regardless of being quarantined at home or back at campus, the perception of parents’ harmonious relationships made students feel acceptance and stability during challenges. However, students whose parents’ relationships were perceived as conflicted became increasingly positive after they returned to campus, indicating that the rise in social mentality could be from sources other than family, which could be investigated in future research. Compared to other parenting styles, the authoritative style was related to a higher social mentality. This is consistent with previous studies’ results [26,55]. As adults, university students’ personalities and values have been formed. Moreover, they have a relatively independent way of thinking and behavior. They spent a long time with their families (parents and siblings) during the home-quarantine phase. Conflicts are likely to occur because of different opinions on living or learning styles. Some students potentially returned to (or remained in) problematic home situations while living under restrictive quarantine conditions, disrupting the identity development processes during the critical life stage of emerging adulthood [56]. The authoritative parenting style made university students feel complete autonomy and abated their negative mentality of university students.

### 4.3. Future Research Directions

As a severe daunting stressor, the COVID-19 pandemic has been dreadful, chronic, and wide-spreading, with serious consequences [51]. The COVID-19 pandemic has brought great challenges to many aspects of people’s lives and has had a profound impact on people’s psychosocial adaptation in many aspects. The clear depiction of the psychosocial adaptation trajectories of different groups under the background of the epidemic has important implications for more effective psychosocial interventions. This longitudinal study of college students’ social mentality fully showed that: (1) social mentality and its fluctuation are very important supplementary evidence to understand the impact of the COVID-19 pandemic; (2) the fluctuation pattern of college students’ social mentality is not quite the same as other social groups or the general social group as a whole; this maybe because the college students have their specific developmental cascades and challenges. Undoubtedly, the timely catching of the social mentality and its perturbations among college students under the COVID-19 pandemic would greatly increase the pertinence and accuracy of mental health services and thus enhance the effectiveness of mental health education from the perspective of early and active intervention.

This study also indicated that for different groups of college students, we can tailor more applicable social mentality guidance programs. For those college students whose social mentality is easily unbalanced, the university should be equipped with more appropriate mental health education resources to improve their psychosocial adaptation through activities or classes so as to reduce the secondary crisis of mental health caused by the COVID-19 epidemic. The feasible approaches include improving positive social mentality, increasing acceptance of change, promoting a sense of hope, boosting coping confidence, and enhancing resilience. We will continue to follow-up with these participants to improve our understanding of how long those outcomes will last. A better understanding of how the COVID-19 affects students’ social mentality can help guide future group interventions among university students.

The results of the optimal mixed-mode suggested that wave, gender, residence, university, grade, being a student cadre, socioeconomic status, parenting styles, and the harmonious degree of parents’ relationships were associated with changing social mentality during the peak and preventive-order phases (see Appendix A). However, it is necessary to further explore the specific mechanisms by which these factors affect social mentality. Regarding the conceptual framework of social mentality, future research could attempt to examine the relationship between individual mental health factors such as anxiety, depression, and stress with various dimensions of social mentality. Based on the strength of positive psychology and resilience, it is worthwhile to further investigate positive social mentality and its associations in public health emergencies, which might be risk perception, view of change, post-traumatic growth, attitude toward vaccination, the estimated end date of the pandemic, interpersonal communication, volunteer service, and awareness of order recovery. Furthermore, it is important to investigate whether resilience-related interventions can improve social mentality under the concept of positive youth development (PYD).

### 4.4. Limitations

From the perspectives of individuals and the public, this study painted a general picture of university students’ social mentality during the pandemic objectively. However, it has several limitations. First, since returning to school was a critical variable, the study determined five nodes (March, April, June, and November 2020 and January 2021) concerning school reopening and the spread. Surveys were conducted every four to eight weeks. However, there is no evidence of a time lag between the mental changes and the situation. Therefore, whether the designed time points are appropriate requires further exploration. It is expected that more follow-up research on university students’ mentality is needed to offer more evidence and conclusions for future research. Second, some respondents who attended the baseline survey could not join the following surveys due to graduation, preparation for final exams, internship, and absence from school. This made the number of those joining the five surveys lower than expected.

## 5. Conclusions

Our findings highlight the changing social mentality among university students during the peak and contained-risk phases of COVID-19, which might help inform other affected regions on how to prepare for the potential increase in mental health problems among university students returning to school. The novelty of this study is its examination of the social mentality of subjects (individual–public) and valence (positive–negative). With the addition of the valence dimension, the investigation of social mentality becomes more revealing and information is more stereoscopic, considering the level and attributes of social mentality.

Positive and negative social mentality coexisted, while overall social mentality first decreased and then increased. Positive facets decreased during the pandemic before increasing in the contained-risk period. It was lowest in June when students began to return to the pandemic-preventive campus from quarantined homes. Students living in provinces (except for Shandong) who were from high-level universities in 2016 and 2017 and majored in medicine displayed a more negative social mentality. Moreover, those who were female, student cadres, non-graduates, and enjoying high socioeconomic status displayed a more positive social mentality. The results of family variables suggested that social mentality among students whose parents’ parenting style was authoritative and who were in the nuclear family was more positive, while the more harmonious relations the parents had, the more positive their children’s social mentality. Waves, gender, residence, university, grade, being a student cadre, socioeconomic status, parenting styles, and harmonious degree of parents’ relationships were considered to be associated with changing social mentality in this study. Further research on the relationship between mental health and social mentality, specifically associations and interventions for positive social mentality, needs to be conducted.

## Figures and Tables

**Figure 1 ijerph-19-03049-f001:**
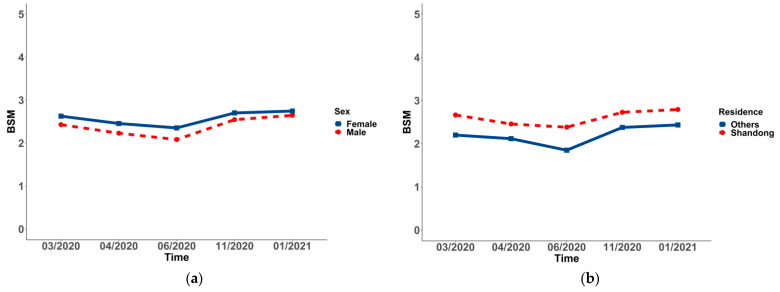
Trends and differences of the overall social mentality (BSM) in gender (**a**) and residence (**b**).

**Figure 2 ijerph-19-03049-f002:**
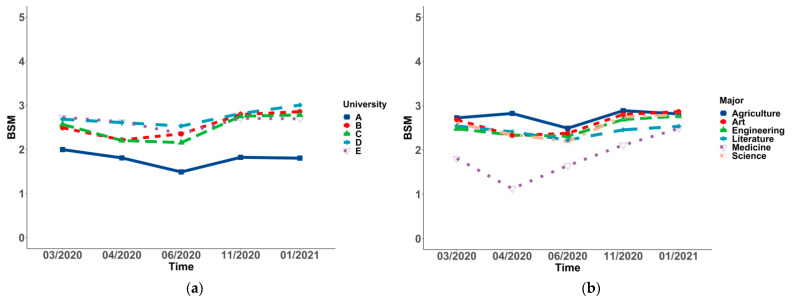
Trends and differences of the overall social mentality (BSM) in university (**a**) and major (**b**).

**Figure 3 ijerph-19-03049-f003:**
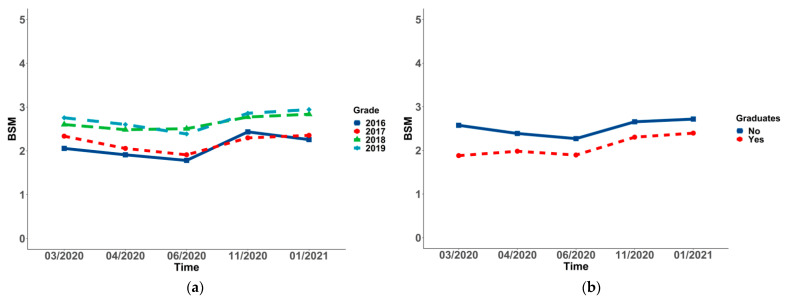
Trends and differences of the overall social mentality (BSM) in grade (**a**) and being a graduate or not (**b**).

**Figure 4 ijerph-19-03049-f004:**
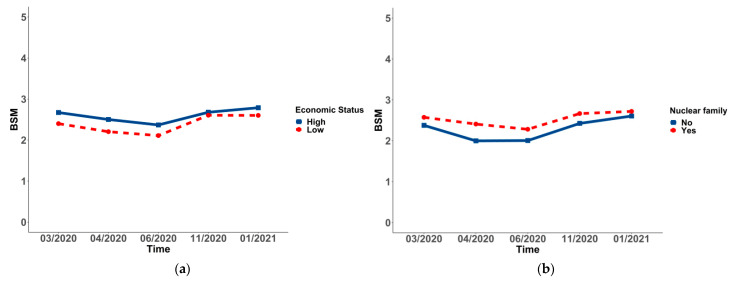
Trends and differences of the overall social mentality (BSM) in social–economic status (**a**) and family style (**b**).

**Figure 5 ijerph-19-03049-f005:**
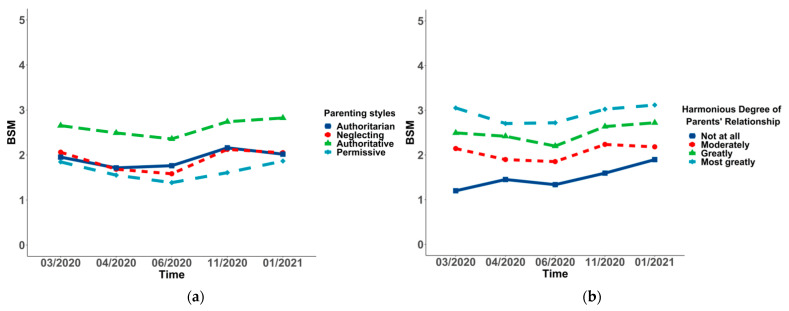
Trends and differences of the overall social mentality (BSM) in parenting style (**a**) and harmonious degree of parents’ relationship (**b**).

**Table 1 ijerph-19-03049-t001:** Response data and Context of the pandemic in Waves 1–5.

Wave	Total Sample (n)	Valid Sample (n)	Recovery Rate (%)	Time	Context of the Pandemic
1	5665	5283	93.26	3–10 March 2020	China was experiencing the full force of the COVID-19 pandemic and all of the university students were home-quarantined
2	5340	4206	78.76	8–15 April 2020	Home-quarantined students were attending online-class
3	4959	4218	85.06	17–24 June 2020	Graduating students returned to campus
4	4832	3962	82.00	1–6 November 2020	Back-to-campus non-graduating students studied in pandemic-preventive order
5	4408	4035	91.54	18–25 January 2021	China’s government started to provide COVID-19 vaccines for all citizens while students went home after completing the autumn-term study on campus
Overlap	1458	1319	90.47		

**Table 2 ijerph-19-03049-t002:** Characteristics of participants at Wave 1 by BSM quantiles (N = 1319).

Variable	Quantile 1 N = 327 Extremely Low	Quantile 2 N = 331 Moderately Low	Quantile 3 N = 329 Moderately High	Quantile 4 N = 332 Extremely High	*p*
Gender					0.150
Male	127 (38.84)	106 (32.02)	111 (33.74)	103 (31.02)	
Female	200 (61.16)	225 (67.98)	218 (66.26)	229 (68.98)	
Ethnic					0.784
Han	318 (97.25)	323 (97.58)	323 (98.18)	322 (96.99)	
Minority	9 (2.75)	8 (2.42)	6 (1.82)	10 (3.01)	
Residence					<0.001
Shandong	226 (69.11)	252 (76.13)	258 (78.42)	283 (85.24)	
Others	101 (30.89)	79 (23.87)	71 (21.58)	49 (14.76)	
University					<0.001
A	68 (20.80)	37 (11.18)	36 (10.94)	21 (6.33)	
B	60 (18.35)	47 (14.20)	66 (20.06)	51 (15.36)	
C	56 (17.13)	63 (19.03)	45 (13.68)	61 (18.37)	
D	68 (20.80)	102 (30.82)	90 (27.36)	94 (28.31)	
E	75 (22.94)	82 (24.77)	92 (27.96)	105 (31.63)	
Major					0.031
Engineering	76 (23.24)	71 (21.45)	83 (25.23)	65 (19.58)	
Science	77 (23.55)	101 (30.51)	91 (27.66)	85 (25.60)	
Agriculture	18 (5.50)	26 (7.85)	21 (6.38)	32 (9.64)	
Literature	111 (33.94)	102 (30.82)	98 (29.79)	102 (30.72)	
Art	33 (10.09)	27 (8.16)	33 (10.03)	45 (13.55)	
Medicine	12 (3.67)	4 (1.21)	3 (0.91)	3 (0.90)	
Grade					<0.001
2016	14 (4.28)	15 (4.53)	5 (1.52)	5 (1.51)	
2017	128 (39.14)	98 (29.61)	83 (25.23)	94 (28.31)	
2018	92 (28.13)	108 (32.63)	111 (33.74)	100 (30.12)	
2019	93 (28.44)	110 (33.23)	130 (39.51)	133 (40.06)	
Student Cadre					0.453
Yes	74 (22.63)	85 (25.68)	89 (27.05)	92 (27.71)	
No	253 (77.37)	246 (74.32)	240 (72.95)	240 (72.29)	
Graduating Student					0.030
Yes	12 (3.67)	11 (3.32)	4 (1.22)	3 (0.90)	
No	31 (9.48)	23 (6.95)	19 (5.78)	23 (6.93)	
Social–economic status					<0.001
High	180 (55.05)	171 (51.66)	208 (63.22)	215 (64.76)	
Low	147 (44.95)	160 (48.34)	121 (36.78)	117 (35.24)	
Nuclear Family					<0.001
Yes	296 (90.52)	308 (93.05)	310 (94.22)	309 (93.07)	
No	31 (9.48)	23 (6.95)	19 (5.78)	23 (6.93)	
Parenting styles					
Authoritarian	42 (12.84)	23 (6.95)	23 (6.99)	11 (3.31)	
Neglecting	25 (7.65)	14 (4.23)	12 (3.65)	10 (3.01)	
Authoritative	253 (77.37)	287 (86.71)	289 (87.84)	309 (93.07)	
Permissive	7 (2.14)	7 (2.11)	5 (1.52)	2 (0.60)	
Harmonious Degree of Parents’ Relationship					<0.001
Not at all	12 (3.67)	4 (1.21)	4 (1.22)	1 (0.30)	
Moderately	89 (27.22)	68 (20.54)	67 (20.36)	39 (11.75)	
Greatly	175 (53.52)	178 (53.78)	164 (49.85)	150 (45.18)	
Most greatly	51 (15.60)	81 (24.47)	94 (28.57)	142 (42.77)	

BSM indicates balanced social mentality.

**Table 3 ijerph-19-03049-t003:** Changes in various dimensions of social mentality (mean ± SD).

Variables	Wave 1 Mar. 2020	Wave 2 Apr. 2020	Wave 3 Jun. 2020	Wave 4 Nov. 2020	Wave 5 Jan. 2021	*p*-Value
PIM	4.60 ± 0.89	4.63 ± 0.92	4.46 ± 1.07	4.71 ± 0.95	4.75 ± 0.98	<0.001 *
NIM	2.01 ± 0.89	2.31 ± 0.97	2.20 ± 1.01	2.12 ± 1.00	2.06 ± 0.98	0.294
PPbM	4.78 ± 0.86	4.76 ± 0.91	4.56 ± 1.04	4.80 ± 0.91	4.82 ± 0.96	0.229
NPbM	2.24 ± 0.95	2.35 ± 1.00	2.30 ± 1.03	2.12 ± 0.98	2.10 ± 0.99	<0.001 *
PSM	4.72 ± 0.79	4.72 ± 0.85	4.53 ± 1.00	4.77 ± 0.87	4.79 ± 0.92	0.011
NSM	2.16 ± 0.83	2.34 ± 0.91	2.27 ± 0.95	2.12 ± 0.93	2.08 ± 0.93	<0.001 *
BSM	2.56 ± 1.43	2.38 ± 1.60	2.26 ± 1.67	2.65 ± 1.64	2.71 ± 1.68	<0.001 *

Data are shown as mean ± SD. *: Adjusted *p* < 0.05 after Bonferroni correction. SD indicates standard deviation; PIM, positive individual mentality; NIM, negative individual mentality; PPbM, positive public mentality; NPbM, negative public mentality; PSM, positive social mentality; NSM, negative social mentality; BSM, balanced social mentality. Analysis of variance was performed to explore the variance of social mentality with time.

**Table 4 ijerph-19-03049-t004:** Post hoc analysis of differences of BSM in Waves 1–5.

BSM	Difference (95% CI)	t	*p*
Wave 2–Wave 1	−0.18 (−0.25, −0.11)	−5.35	<0.001 *
Wave 3–Wave 1	−0.30 (−0.37, −0.22)	−8.01	<0.001 *
Wave 4–Wave 1	0.09 (−0.03, 0.21)	1.47	0.142
Wave 5–Wave 1	0.15 (0.03, 0.27)	2.52	0.012
Wave 3–Wave 2	−0.12 (−0.19, −0.05)	−3.23	0.001 *
Wave 4–Wave 3	0.39 (0.26, 0.51)	6.00	<0.001 *
Wave 5–Wave 4	0.06 (−0.07, 0.19)	0.96	0.338

*: Adjusted *p* < 0.05 after Bonferroni correction. Paired *t*-tests were used for the difference of BSM between waves. BSM, balanced social mentality; CI, confidence interval.

**Table 5 ijerph-19-03049-t005:** Changes and differences of BSM in socio-demographic variables (mean ± SD).

Variables	Wave 1	Wave 2	Wave 3	Wave 4	Wave 5	*p for*	*p for*
	Mar. 2020	Apr. 2020	Jun. 2020	Nov. 2020	Jan. 2021	Group	Time
Gender						<0.001 *	<0.001
Female	2.63 ± 1.38	2.46 ± 1.53	2.35 ± 1.59	2.70 ± 1.56	2.74 ± 1.64		
Male	2.43 ± 1.52	2.23 ± 1.71	2.08 ± 1.80	2.54 ± 1.78	2.65 ± 1.77		
Ethnicity						0.761	<0.001
Han	2.56 ± 1.43	2.38 ± 1.60	2.26 ± 1.66	2.65 ± 1.64	2.70 ± 1.68		
Minority	2.51 ± 1.54	2.39 ± 1.63	2.33 ± 1.79	2.54 ± 1.67	2.97 ± 1.73		
Residence						<0.001 *	<0.001
Shandong	2.67 ± 1.42	2.46 ± 1.60	2.38 ± 1.64	2.73 ± 1.63	2.79 ± 1.66		
Others	2.20 ± 1.43	2.12 ± 1.57	1.85 ± 1.68	2.38 ± 1.66	2.43 ± 1.75		
University						<0.001 *	<0.001
A	2.00 ± 1.53	1.81 ± 1.65	1.49 ± 1.41	1.82 ± 1.50	1.80 ± 1.54		
B	2.49 ± 1.49	2.22 ± 1.62	2.35 ± 1.69	2.79 ± 1.70	2.86 ± 1.73		
C	2.57 ± 1.39	2.20 ± 1.50	2.15 ± 1.64	2.75 ± 1.62	2.78 ± 1.62		
D	2.68 ± 1.31	2.61 ± 1.48	2.53 ± 1.60	2.81 ± 1.55	3.01 ± 1.55		
E	2.73 ± 1.44	2.63 ± 1.65	2.36 ± 1.73	2.71 ± 1.66	2.69 ± 1.76		
Major						<0.001 *	<0.001
Agriculture	2.73 ± 1.46	2.83 ± 1.58	2.49 ± 1.62	2.89 ± 1.55	2.82 ± 1.85		
Art	2.68 ± 1.40	2.33 ± 1.63	2.38 ± 1.75	2.80 ± 1.72	2.87 ± 1.68		
Engineering	2.48 ± 1.47	2.34 ± 1.64	2.31 ± 1.67	2.69 ± 1.68	2.76 ± 1.69		
Literature	2.54 ± 1.43	2.41 ± 1.54	2.22 ± 1.71	2.46 ± 1.65	2.54 ± 1.73		
Medicine	1.81 ± 1.37	1.13 ± 2.04	1.65 ± 1.46	2.11 ± 1.60	2.48 ± 1.56		
Science	2.60 ± 1.41	2.35 ± 1.55	2.19 ± 1.60	2.75 ± 1.56	2.80 ± 1.58		
Grade						<0.001 *	<0.001
2016	2.06 ± 1.33	1.91 ± 1.56	1.78 ± 1.62	2.43 ± 1.57	2.26 ± 1.63		
2017	2.34 ± 1.53	2.06 ± 1.64	1.91 ± 1.68	2.30 ± 1.68	2.36 ± 1.72		
2018	2.60 ± 1.36	2.49 ± 1.51	2.51 ± 1.57	2.77 ± 1.61	2.84 ± 1.64		
2019	2.76 ± 1.39	2.60 ± 1.60	2.39 ± 1.69	2.86 ± 1.59	2.94 ± 1.65		
Student cadre						<0.001*	<0.001
Yes	2.68 ± 1.38	2.60 ± 1.52	2.37 ± 1.63	2.78 ± 1.7	2.88 ± 1.66		
No	2.52 ± 1.45	2.30 ± 1.62	2.22 ± 1.68	2.60 ± 1.61	2.65 ± 1.69		
Graduates						0.001 *	<0.001
Yes	1.88 ± 1.35	1.98 ± 1.63	1.89 ± 1.66	2.30 ± 1.68	2.40 ± 1.71		
No	2.58 ± 1.43	2.39 ± 1.60	2.27 ± 1.67	2.66 ± 1.64	2.72 ± 1.68		
Nuclear family						0.001 *	<0.001
Yes	2.57 ± 1.43	2.41 ± 1.57	2.28 ± 1.66	2.67 ± 1.64	2.72 ± 1.69		
No	2.38 ± 1.49	2.00 ± 1.89	2.01 ± 1.69	2.43 ± 1.67	2.60 ± 1.60		
Social Economic Status						<0.001 *	<0.001
High	2.67 ± 1.44	2.50 ± 1.59	2.37 ± 1.69	2.68 ± 1.61	2.79 ± 1.70		
Low	2.40 ± 1.41	2.20 ± 1.59	2.11 ± 1.62	2.60 ± 1.68	2.60 ± 1.65		
Parenting styles						<0.001 *	<0.001
Authoritarian	1.95 ± 1.44	1.71 ± 1.49	1.76 ± 1.54	2.16 ± 1.57	2.02 ± 1.68		
Neglecting	2.06 ± 1.47	1.68 ± 1.91	1.58 ± 1.81	2.12 ± 1.66	2.05 ± 1.46		
Authoritative	2.65 ± 1.41	2.49 ± 1.57	2.36 ± 1.65	2.74 ± 1.63	2.82 ± 1.67		
Permissive	1.85 ± 1.51	1.55 ± 1.44	1.39 ± 1.63	1.61 ± 1.62	1.87 ± 1.90		
Harmonious Degree of Parents’ Relationship						<0.001 *	<0.001
Not at all	1.20 ± 1.62	1.45 ± 1.49	1.34 ± 1.94	1.59 ± 1.82	1.90 ± 1.70		
Moderately	2.14 ± 1.41	1.90 ± 1.57	1.85 ± 1.65	2.24 ± 1.57	2.18 ± 1.59		
Greatly	2.49 ± 1.41	2.42 ± 1.55	2.20 ± 1.62	2.64 ± 1.65	2.72 ± 1.69		
Most greatly	3.05 ± 1.31	2.70 ± 1.61	2.72 ± 1.63	3.02 ± 1.57	3.12 ± 1.62		

*: Adjusted *p* < 0.05 after Bonferroni correction. Data were shown as mean ± SD. BSM indicates overall social mentality; SD, standard deviation. Repeated measures analysis of variance was performed to explore the variance of social mentality with time and groups of socio-demographic variables.

## Data Availability

The questionnaire used in the study and datasets can be made available upon request from the corresponding author.

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
