# Peer review of "Changing Social Mentality among University Students in the COVID-19 Pandemic: A Five-Wave Longitudinal Study in China"

_ijerph, 2022, doi:10.3390/ijerph19053049_

Round 1

Reviewer 1 Report

Review Manuscript IJERPH- 1596782

Dear author(s),

First of all, thank you for the opportunity to review this manuscript. Please, see below some suggestions for your evaluation if they can contribute to improve your work.

Abstract

The abstract is well written and contains the main elements of the paper. The chosen keywords complement the title and abstract, contributing to the study being found by researchers interested in the topic.

Introduction

The introduction is well-structured and provides the necessary elements for the reader to understand the research gap and the aim of the authors. The authors show the importance of the theme in an adequate and forceful way, citing appropriate references.

Please consider commenting on how your research complements and is complemented by the research by Li et al. (2020). As one of the few references of the longitudinal type, this comment would be valuable to emphasize the contribution of the study and for future ones.

Materials and methods

The method is well explained, in great detail, and is suitable for both the proposed objectives and the existing research gap.

Please verify the repetition of the sentence “Each IP address was allowed only one questionnaire response” in the lines 135 and 140

Please consider clarifying how a response was defined as illogical (line 142)

The methodological procedures and the moments in time chosen for the waves are described objectively and clearly. As a suggestion to make it more visual, a table could be created summarizing the following data: wave / month and corresponding explanation of the context of the pandemic / number of responses obtained / recovery rate

Line 182 / 184 – Please consider writing the number (three questions, one meter) so that the text is not confused with the scales (4-point Likert, 5-point), scores and questionnaire names (DASS-21)

Please consider the same for line 206 where you wrote “5 different universities who had participated in all five waves”

Please consider the same for lines 260 (two wave study) and 262 (4-wave study); consider following a pattern throughout the text

Line 189 – Missing space before “Informed”

Results and discussion

Please check if it is possible to increase the size of the words/data in the graphics in Figure 1. It is possible to understand using the zoom, but if this can be avoided, it would enhance data presentation and help in the fluidity of the reading and, consequently, in the pleasantness and understanding of the text.

Please check typo in line 318 (dngerously)

From line 354 to 361 you have commented on regional differences. This is very interesting and yet very little is known about it. Please also consider commenting on the need to consider populations from underdeveloped countries, since they do not have adequate economic, social and educational conditions, which affects the ability to stop the advance and fight against covid-19.

Please consider providing more concrete suggestions for future research.

Thanks again for the opportunity to review this work.

Author Response

Dear Editor/Reviewers,

We would like to express our appreciation to the editor for arranging the review and the valuable comments provided by the reviewers. We have carefully considered and discussed your comments point by point, and made careful corrections and modifications to the article, all of which have been highlighted. Our specific responses are as follows. After revision, paragraphs and lines have been changed a lot. Its noted that Lines... here refers to the revised manuscript we submitted using the tracking pattern in WPS.

Responses to Reviewer 1

Introduction

  • Please consider commenting on how your research complements and is complemented by the research by Li et al. (2020). As one of the few references of the longitudinal type, this comment would be valuable to emphasize the contribution of the study and for future ones.

Response:Thanks for addressing this point. As suggested by the reviewer, we have added some comments on this article to emphasize our study’s contribution in Lines 202-213 in the Introduction section.

Materials and methods 

  • Please verify the repetition of the sentence “Each IP address was allowed only one questionnaire response” in the lines 135 and 140.

Response: Thanks a lot for noting the repetition of the sentence “Each IP address was allowed only one questionnaire response.” We checked it carefully and deleted the second repeated sentence “ Each IP address was allowed one questionnaire response” in Lines 297-298.

  • Please consider clarifying how a response was defined as illogical (line 142)

Response: As suggested by the reviewer, we added an explanation sentence in Lines 300-303. One trap question (e.g. Select “Satisfied” in this question to check the level of careful reading ) in each survey was set up to check if the respondent was reading carefully. Responses choosing the wrong answer was illogical. Furthermore, the trap questions in each wave are shown in the table below.

Wave

Trap Question

Option

Logical answer

Illogical answer

1

Please note that the following two questions require you to choose between a personal and a public perspective. Do you understand the meaning of the question?

Yes/No

Yes

No

2

Please note that the following two questions require you to choose between a personal and a public perspective. Do you understand the meaning of the question?

Yes/No

Yes

No

3

Please note that the following two questions require you to choose between a personal and a public perspective. Do you understand the meaning of the question?

Yes/No

Yes

No

4

Select “Satisfied” in this question to check the level of careful reading

Satisfied/Dissatisfied

Satisfied

Dissatisfied

5

Select “1” in this question to check the level of careful reading

1/2/3

1

2/3

  • The methodological procedures and the moments in time chosen for the waves are described objectively and clearly. As a suggestion to make it more visual, a table could be created summarizing the following data: wave / month and corresponding explanation of the context of the pandemic / number of responses obtained / recovery rate

Response: Many thanks for addressing this. As suggested by the reviewer, we have deleted words in Lines 248-255, and added words (Lines 246-247) and Table 1 as follows in Lines 256-268 to summarize specific data in each wave.

Wave

Total sample 

(n)

Valid sample

(n)

Recovery rate

(%)

Time

Context of the pandemic

1

5665

5283

93.26

March 3-10, 2020

China was experiencing the full force of the COVID-19 pandemic and all of the university students were home-quarantined

2

5340

4206

 78.76

April 8-15, 2020

Home-quarantined students was attending online-class

3

4959

4218

85.06

June 17-24, 2020

Graduating students returned to campus

4

4832

3962

82.00

November 1-6, 2020

Back-to-campus non-graduating students studied in pandemic-preventive order

5

4408

4035

91.54

January 18-25, 2021

China government started to provide COVID-19 vaccine for all, while students went home after completing the autumn-term study on campus

Overlap

1458

1319

90.47

  • Line 182 / 184 – Please consider writing the number (three questions, one meter) so that the text is not confused with the scales (4-point Likert, 5-point), scores and questionnaire names (DASS-21)

Response: Thanks for noting this.We have check it carefully and added explanation on the measurement. Regarding the aim of this study, we have deleted the psychological variables such as DASS-21 etc. in Lines 341-361.

  • Please consider the same for line 206 where you wrote “5 different universities who had participated in all five waves”

Response: Thanks for noting this.We have check it carefully and revised it in Line 240, changing 30 for thirty and 5 for five.

  • Please consider the same for lines 260 (two wave study) and 262 (4-wave study); consider following a pattern throughout the text

Response: Thanks for noting this.We have check it carefully and made modifications in a unified pattern in the text, specifically in Lines 214 (changing 5-wave for five-wave). Furthermore, we use waves 1-5 or waves 1 to 5 to follow a unified pattern throughout the text.

  • Line 189 – Missing space before“Informed”

Response: Thanks lot for noting this. We have added a space before “Informed” in Lines 364-365.

Results and discussion

  • Please check if it is possible to increase the size of the words/data in the graphics in Figure 1. It is possible to understand using the zoom, but if this can be avoided, it would enhance data presentation and help in the fluidity of the reading and, consequently, in the pleasantness and understanding of the text.

Response: Thanks a lot for addressing this. As requested by the reviewer, we have checked it carefully and made major modification to improve the pleasantness and understanding of the text. Firstly, we find it difficult to increase the size of the data in the graphics in Figure 1 because of the limited zoom, so Figure 1 in Lines 511-512 is changed for five specific figures in Lines 498-508 to enhance data presentation. Secondly, we added more words in Lines 446-467 to further interpret the meaning in figures presented above. Furthermore, we transferred Supplement Appendix S1 (characteristics of the sample in baseline survey) to Table 2 ( Lines 401-402) in the Result section in order to be more readable.

  • Please check typo in line 318 (dngerously)

Response: Thanks lot for noting this. As requested by another reviewer, we have carefully checked it and deleted this paragraph in Lines 638-649 for considering the content as political and unrelated with the results in this study.

  • From line 354 to 361 you have commented on regional differences. This is very interesting and yet very little is known about it. Please also consider commenting on the need to consider populations from underdeveloped countries, since they do not have adequate economic, social and educational conditions, which affects the ability to stop the advance and fight against covid-19.

Response: Thanks a lot for noting this point. As suggested by the reviewer, we have added more words to discuss on regional difference in Lines 700-706.

  • Please consider providing more concrete suggestions for future research.

Response: As requested by the reviewer, we have added more words and summed up in three paragraphs in 4.3 for depicting more concrete suggestions for future research in the text in Lines 784-826. The text is as follows.

As a severe daunting stressor, COVID-19 pandemic is dreadful, chronic, wide spreading with serious consequences. COVID-19 pandemic has brought great challenges to many aspects of people's life and had a profound impact on people's psychosocial adaptation in many aspects. The clear depicting of the psychosocial adaptation trajectories of different groups under the background of the epidemic has important implications for more effective psychosocial interventions. This longitudinal study of college students' social mentality fully showed that: 1) social mentality and its fluctuation are very important supplementary evidence to understand the impact of COVID-19 pandemic; 2) the fluctuation pattern of college students’ social mentality is not quite the same as of other social group or the general social group as a whole, this maybe because the college students have their specific developmental cascades and challenges. Undoubtedly, timely catching of the social mentality and its perturbations among college students under COVID-19 pandemic would greatly increase the pertinence and accuracy of mental health services and thus enhance the effectiveness of mental health education from the perspective of early and active intervention.

This study also indicated that for different groups of college students, we can tailor more applicable social mentality guidance programs. For those college students whose social mentality is easy to be unbalanced, the university should be equipped with more appropriate mental health education resources to improve their psychosocial adaptation through activities or classes, so as to reduce the secondary crisis of mental healthcaused by COVID-19 epidemic. The feasible approaches include improving positive social mentality, increasing acceptance of change, enhancing a sense of hope, boosting coping confidence and enhancing resilience. We will continue to follow up these participants to improve our understanding about how long those outcomes will last. A better understanding of how the COVID-19 affects students' social mentality can help guide future group interventions among university students.

The results of the optimal mixed-mode suggested that wave, gender, residence, university, grade, whether being a student cadre, socioeconomic status, parenting styles, and harmonious degree of parents’ relationships were associated with changing social mentality during the peak and preventive-order phases (see Supplementary Table S2). However, it is necessary to further explore the specific mechanisms by which these factors affect social mentality. Regarding the conceptual framework of social mentality, future research could attempt to examine the relationship between individual mental health factors such as anxiety, depression, and stress with various dimensions of social mentality. Based on the strength of positive psychology and resilience, it is worthwhile to further investigate positive social mentality and its associations in public health emergencies, which might be risk perception, view of change, post-traumatic growth, attitude toward vaccination, the estimated end date of the pandemic, interpersonal communication, volunteer service, and awareness of order recovery. Furthermore, it is important to investigate whether resilience-related interventions can improve social mentality under the concept of positive youth development (PYD).

  • Language and style in the revised manuscript have been polished by a native speaker in English.

Reviewer 2 Report

Some information from Introduction should be transferred to Discussion, because authors instead of showing the background of the study they refer to other such studies. They should expand on these references and discuss in section Discussion.

Please explain how the sample was randomized - if You send questionare online.

Figure 1 contains too many graphs, which are completely unreadable.

Data from Figure 1 are not interpreted in the results section.

Conclusion is too short and conclusions are too general.

Author Response

Dear Editor/Reviewers,

We would like to express our appreciation to the editor for arranging the review and the valuable comments provided by the reviewers. We have carefully considered and discussed your comments point by point, and made careful corrections and modifications to the article, all of which have been highlighted. Our specific responses are as follows. After revision, paragraphs and lines have been changed a lot. Its noted that Lines... here refers to the original revision using the tracking pattern in WPS. 

Responses to Reviewer 2

  • Some information from Introduction should be transferred to Discussion, because authors instead of showing the background of the study they refer to other such studies. They should expand on these references and discuss in section Discussion.

Response: Many thanks for the comment and suggestion. As requested by the reviewer, we revised manuscript in Lines 518-519, 529-530, 672, 716 in which some information from Introduction was transferred to Discussion.

  • Please explain how the sample was randomized - if You send questionnaire online.

Response: Thank you very much for your comment. In this study, students of thirty majors from five universities in Shandong province were recruited through stratified cluster random sampling. Firstly, we randomly selected five universities from Shandong Province, including one comprehensive top university and four ordinary universities. Secondly, we randomly selected 6 majors from all majors in each university. Thirdly, one class from each grade of each major was randomly selected respectively. The randomization was performed in the method of random number. We have added specific explanation in Lines 190-193, that is, “Firstly...Thirdly” in the text.

  • Figure 1 contains too many graphs, which are completely unreadable.

Response: Thanks very much for the comment. In order to make it more readable, we made major modification in this part. Figure 1 in Lines 513-514 was changed for five specific figures in Lines 374-384 to enhance data presentation.

  • Data from Figure 1 are not interpreted in the results section.

Response: Thanks very much for the comment. We added more words in Lines 331-351 to further interpret the meaning in figure 1-5 in Lines 374-384

  • Conclusion is too short and conclusions are too general.

Response: As a severe daunting stressor, COVID-19 pandemic is dreadful, chronic, wide spreading with serious consequences. It has brought great challenges to many aspects of people's life and had a profound impact on people's psychosocial adaptation in many aspects. The clear depicting of the psychosocial adaptation trajectories of different groups under the background of the epidemic has important implications for more effective psychosocial interventions. This longitudinal study of college students' social mentality fully showed that: 1) social mentality and its fluctuation are very important supplementary evidence to understand the impact of COVID-19 pandemic; 2) the fluctuation pattern of college students’ social mentality is not quite the same as of other social group or the general social group as a whole, this maybe because the college students have their specific developmental cascades and challenges. Undoubtedly, timely catching of the social mentality and its perturbations among college students under COVID-19 pandemic would greatly increase the pertinence and accuracy of mental health services and thus enhance the effectiveness of mental health education from the perspective of early and active intervention.

Many thanks for the comment. We have checked it carefully and made considerable modification in Lines 605-628. We concluded in three points: the novelty of this study, the specific changing trajectories and difference of social mentality among university students and suggestions on future research. The text is as follows.

Our findings highlight the changing social mentality among university students during the peak and contained-risk phases of COVID-19, which might help inform other affected regions affected how to prepare for the potential increase in mental health problems among university students returning to school. The novelty of this study is its examination of the social mentality of subjects (individual–public) and valence (positive–negative). With the addition of the valence dimension, the investigation of social mentality becomes more revealing, and information is more stereoscopic, considering the level and attributes of social mentality.

Positive and negative social mentality coexisted, while overall social mentality first decreased and then increased. Positive facets decreased during the pandemic before increasing in the contained-risk period. It was lowest in June when students began to return to the pandemic-preventive campus from quarantined homes. Students living in provinces except for Shandong, who were from high-level universities in 2016 and 2017 and majored in medicine, displayed a more negative social mentality. Moreover, those who were female, student cadres, non-graduates, and enjoying high socioeconomic status displayed a more positive social mentality. The results of family variables suggested that social mentality among students whose parents’ parenting style was authoritative and who were in the nuclear family was more positive, while the more harmonious relations parents had, the more positive their children’s social mentality. Waves, gender, residence, university, grade, whether being a student cadre, socioeconomic status, parenting styles, and harmonious degree of parents’ relationships were considered to be associated with changing social mentality in this study. Further research on the relationship between mental health and social mentality, specifically associates and interventions for positive social mentality, needs to be conducted.

Language and style in the revised manuscripthave been polished by a native speaker in English.

Reviewer 3 Report

This paper reports new longitudinal data on the effects of the COVID pandemic on student mental health (I think, see below).  I have observations as follows (in no particular order):

1)   The term ‘social mentality’ is used to indicate related but slightly different constructs within social science. Social mentality theory focusses on care-giving and care seeking (Hurmanto and Zuroff, 2015) while Li (2014) uses the term to mean “Social mentality presents the “barometer” of change and development of a society. Special historical background and practical reasons contributed to, among our citizens, the social mentality of anxiety, an unbalanced one, the essential reason for which is the surpass of conflict between human inner needs and social reality over human adaptive ability”.  I am assuming the authors are using it in terms of the latter, but this I believe is a China-specific concept that might be worth explaining in more detail for the benefit of the non-Chinese audience.   Fundamentally I was not sure what it ‘mentality’ was.

Is social mentality an individual difference variable akin to the mental well-being of the individual or is it an indicator of the state of a group of people/society as indicated in Li’s quote above?  The description presented suggests it is something to do with the latter but the measurement section of the paper suggests that it is operationalised as a collection of individual difference measures. What does ‘positive (or negative) mentality’ mean? What is ‘mentality’? Is it orientation? Anxiety?  Later on in lines 254-255 ‘mentality’ appears to mean a range of conventional individual mental health indicators. If so, why appeal to the concept of ‘social mentality’, why not just ‘mental health’?  I looked for the BDSMQ online but could only find Chinese language sources so it would be helpful to have some English translations of the survey items so that we can understand what the various BDSMQ items are tapping into.  Without this it is difficult to work out what the paper is demonstrating.  It would also help us to see how this differed from the other measures used like the DASS and Sense of Security scale.

2) The response rates are extremely high compared with most Western longitudinal social surveys of this sort. Can the authors say something about the social context in which the study was done since figures this high in the West would suggest coercion or some influence that might bring into question the validity of the responses.  The section saying “Full-time tutors who majored in psychology, education, and administration were trained online, including the study protocol, personal information, data collection procedures, and secure personal information to help the students during the survey” suggests, from a Western perspective, that the respondents may have been under some implicit pressure to complete the surveys and/or to provide socially acceptable responses.

3) While the authors have used null hypothesis significance testing (NHST) approaches the paper is essentially a descriptive one. The aim is to document the change in the key variables during a very specific time in history and there are no formal hypotheses being tested here (are there?).  The p-values associated with the various tests are being used as descriptive rather than inferential statistics and it might be better to simply present the data with confidence intervals and omit the NHST testing.  If the tests are to be retained then some form of Type 1 error protection (e.g. Bonferroni-Holm correction) would be advisable.  It would also be appropriate to report indicators of effect sizes since many of the effects in the graphs in Figure 1 appear to be quite modest. 

4) The material in lines 318 -328 seems to be a political statement in support of the Chinese government and seems unrelated to the data presented. Similarly on lines 331 and onwards it is not clear what the Chinese policy of placing vaccines on the international market has to do with the promotion of individual mental resilience in students. Lines 391-399 contain more political polemic.  Whether the actions of the Chinese authorities worked well or not will be a matter for historians to decide, the present data say nothing about this.

5) On line 388 it says “Negative mentality (anxiety, depression, pressure, worry, information-burnout, etc.) was constantly relieved, and positive mentality (peacefulness, serenity, gratitude, happiness, etc.) was constantly reinforced. The optimal mixed mode in this study (see Supplementary Table S2).” But this is not what the data shows as there is first a decline followed by a mild recovery. The graphs in Figure 1 show relatively minor changes across time.

6) Line 406 discusses parenting styles but this was not explained in the method or results sections.

Author Response

Dear Editor/Reviewers,

We would like to express our appreciation to the editor for arranging the review and the valuable comments provided by the reviewers. We have carefully considered and discussed your comments point by point, and made careful corrections and modifications to the article, all of which have been highlighted. Our specific responses are as follows. After revision, paragraphs and lines have been changed a lot. Its noted that Lines... here refers to the original revision using the tracking pattern in WPS. 

Responses to Reviewer 3

Introduction

  • Explanations on the term social mentality and BDSMQ

  The term ‘social mentality’ is used to indicate related but slightly different constructs within social science. Social mentality theory focusses on care-giving and care seeking (Hurmanto and Zuroff, 2015) while Li (2014) uses the term to mean “Social mentality presents the “barometer” of change and development of a society. Special historical background and practical reasons contributed to, among our citizens, the social mentality of anxiety, an unbalanced one, the essential reason for which is the surpass of conflict between human inner needs and social reality over human adaptive ability”.  I am assuming the authors are using it in terms of the latter, but this I believe is a China-specific concept that might be worth explaining in more detail for the benefit of the non-Chinese audience.   Fundamentally I was not sure what it ‘mentality’ was.

Is social mentality an individual difference variable akin to the mental well-being of the individual or is it an indicator of the state of a group of people/society as indicated in Li’s quote above?  The description presented suggests it is something to do with the latter but the measurement section of the paper suggests that it is operationalised as a collection of individual difference measures. What does ‘positive (or negative) mentality’ mean? What is ‘mentality’? Is it orientation? Anxiety?  Later on in lines 254-255 ‘mentality’ appears to mean a range of conventional individual mental health indicators. If so, why appeal to the concept of ‘social mentality’, why not just ‘mental health’?  I looked for the BDSMQ online but could only find Chinese language sources so it would be helpful to have some English translations of the survey items so that we can understand what the various BDSMQ items are tapping into.  Without this it is difficult to work out what the paper is demonstrating.  It would also help us to see how this differed from the other measures used like the DASS and Sense of Security scale.

Response:Thank you very much for the comment and suggestion. Three parts are included in our responses.

  1. The concept of social mentality.

You’re correct. Social mentality in this study is different from that in social mentality theory (Hurmanto and Zuroff, 2015) but familiar with that in the study by Li. (2014) you referred to in your comment. Although the concept of social mentality in this study is China-specific, it arises from the fields of history, philosophy, and social psychology. The term "mentality" was coined in the academic field of historiography. There are two primary definitions in Larus' French Dictionary: one is the spiritual customs, beliefs, and emotions of a group, and the other is the individual mental state (or psychological state) (Lu Yimin, 1992). The term "mentality" originated in British philosophy and refers to the psychological characteristics shared by a nation or a human group, expressed in sensory and mental ways.

Research on social mentality in the field of social psychology begins with two major directions. Social psychology in North America is primarily concerned with the process by which individuals are passively influenced by others (that is, how groups actively influence individuals) (Yu Guoliang, Wei Qingwang, 2014). Additionally, European social psychology examines individuals' active integration into groups (Fang Wen, 2002). Yang Yiyin (2006) from China combined the two and proposed a ring structure from individual to group, namely the interactive construction process model including individual psychology and social mentality. The model is a dynamic, active system in which social mentality is not only influence variable, but also process and outcome variable. The model is aimed to reveal mutual construction of the individual and society in the most macro psycho-social relation. According to social psychology research, we believe that social mentality in this study refers to the cognition, emotions, values, and behaviors permeating the whole society or social group/social category in a period of time. Specifically, it displays the general state of mind in the group-level, interactively constructed by individual and society, with the characteristics of changeability, universality, simplicity, emotionality, and infectiousness. It’s not simple combination of numerous individual specific responses, but constructed among individuals by social identification to exhibit similar mood in the group-level.

  1. Significance of exploring social mentality among university students in this study.

Since the outburst of the pandemic, most of research have been focused on mental health. Although both mental health and social mentality are evaluated using individual indicators, the latter evaluates the psychological state of the individual in addition to the psychological state of the group from the perspective of group members, which reflects the perspective of construction of individual and public. According to the interaction model of social mentality by Yang (2006) (which has been introduced before), individual psychology and social mentality interact in the context of COVID-19. Individuals’ anxiety, depression, and stress are influenced by interpersonal interaction and group identification. In other words, individual state of mind is influenced by group values and attitudes. Therefore, based on research on mental health, it is necessary to investigate the psychological state in group level in order to improve group intervention and guidance, create a positive social atmosphere, and promote social stability and harmony. Many researchers in China explored social mentality among specific groups such as young adults, healthcare workers and so on (See References section). Therefore, this study is formulated based on interpretation above and the aim of this study is to explore the social mentality among university students which is a vulnerable group.

The clear depicting of the psycho-social adaptation trajectories of different groups under the background of the epidemic has important implications for more effective psycho-social interventions. This longitudinal study of college students' social mentality fully showed that: 1) social mentality and its fluctuation are very important supplementary evidence to understand the impact of COVID-19 pandemic; 2) the fluctuation pattern of college students’ social mentality is not quite the same as of other social group or the general social group as a whole, this maybe because the college students have their specific developmental cascades and challenges. Undoubtedly, timely catching of the social mentality and its perturbations among college students under COVID-19 pandemic would greatly increase the pertinence and accuracy of mental health services and thus enhance the effectiveness of mental health education from the perspective of early and active intervention.

  • Measurement of social mentality in this study.

Questionnaires are widely used to assess social mentality in multi-level dimensions and descriptive analysis is frequently applied for statistical analysis (Wang J.X. et al., 2020). However, due to the diversity of individual reactions and the heterogenicity of samples, it is difficult to get a comprehensive knowledge of social mentality in depth. We believe that social mentality should be investigated in both macro and micro level. As a new trend, resilience psychology and positive psychology seek to discover and stimulate people's healthy life vitality as a personal responsibility, to investigate how people actively participate successfully in social change and maintain a healthy development function, thereby providing research on social mentality with a promising paradigm for reference.  Furthermore, it’s particularly important to examine positive adjustment among young adults during the public health emergency such as COVID-19 pandemic. Therefore, the conceptual framework design for this study incorporates two considerations and breakthroughs.

First, while considering the macro-micro level, we also consider “positive-negative” mentality, where one dimension is “individual-public” and the other is “positive-negative”. The former is the subject dimension of social mentality, encompassing both micro and macro; the latter is the valence dimension of social mentality, encompassing both negative and positive.

Second, just as positive psychology introduced the emotional balance index when examining the emotional dimension of happiness (Derogatis & Rutigliano, 1996), we believe that the social mentality balance index should be introduced when examining the characteristics of social mentality. The emotional balance index is calculated by subtracting the difference between positive and negative emotion, i.e. BA=PA-NA, where BA represents the emotional balance index score, PA represents positive emotion, and NA represents negative emotion. The emotional balance index measures the balance of positive and negative emotion. Similarly, we can define the social mentality balance index, as expressed by the formula: BSM=PSM-NSM, where BSM denotes the social mentality balance index, PSM denotes the positive social mentality score, and NSM denotes the negative social mentality score. The addition of a social mentality balance index will enliven and enrich investigation of social mentality. Simultaneously, it effectively accounts for two distinct social mentalities and makes social mentality more integrated.

Furthermore, the Bi-Dimensional Structure Questionnaire of Social Mentality (BDSMQ) we used to measure social mentality is designed in two dimensions, where one dimension is “individual-public” and the other is “positive-negative”. The former is the subject dimension of social mentality, encompassing both micro and macro; the latter is the valence dimension of social mentality, encompassing both negative and positive. The scale contains 46 words or phrases(21 for positive social mentality and 25 for negative social mentality). Meanwhile, 25 of those phrases above are used for the individual mentality and 34 for the public mentality. The BDSMQ includes positive individual mentality (PIM), negative individual mentality (NIM), positive public mentality (PPbM), negative public mentality (NPbM), positive social mentality (PSM), negative social mentality (NSM), and balanced social mentality (BSM). PSM is comprised of PIM and PPbM, while NSM is comprised of NIM and NPbM. BSM, which is the balance between PSM and NSM, that is, BSM = PSM-NSM, indicates the proportion of positive social mentality and describes the overall social mentality based on the subject (individual-public) and valence (positive-negative). The scale applied the item presentation method in PANAS ( Positive and Negative Affect Schedule, a widely used scale evaluating emotions), displaying specific phrases of social mentality such as being insecure, being hopeful, being faithful, being tolerant, being harmonious, being unfair, being honest, being grateful, being supportive, etc.. The subjects were asked to rate these items from their standpoint and the public's point of view. Responses were rated on a 6-point Likert scale ranging from 1 (not at all) to 6 (mostly). Higher BSM indicates that the overall social mentality is more positive. The validity and reliability have been confirmed in previous studies (Xi J. Z., 2019). The Cronbach's of BDSMQ in this study ranged from 0.917 to 0.970.

In order to make the concept and measurement of social mentality clearer and more readable, we have made considerable modification in the text. Firstly, we added five paragraphs labeled 1.1 (Lines 57-114) in Introduction section to explain what ‘social mentality’ is and how to evaluate it in this study based on literature review. Secondly, we added more words in detail to interpret the BDSMQ in Lines 248-255 and Lines 261-266 (See 2.3.2 in Materials and Methods section). Furthermore, we have deleted the measurement of psychological variables in Lines 338-358 regarding the aim of this study.

  • The response rates are extremely high compared with most Western longitudinal social surveys of this sort. Can the authors say something about the social context in which the study was done since figures this high in the West would suggest coercion or some influence that might bring into question the validity of the responses.  The section saying “Full-time tutors who majored in psychology, education, and administration were trained online, including the study protocol, personal information, data collection procedures, and secure personal information to help the students during the survey” suggests, from a Western perspective, that the respondents may have been under some implicit pressure to complete the surveys and/or to provide socially acceptable responses.

Response: Thank you very much for comment. We feel sorry that we did not provide enough information about the process of university students recruitment. In fact, respondents were totally voluntary recruitment. In China, tutors hold class meetings during the pandemic online because of the social distancing policy. Therefore, a link to the questionnaire was sent to potential respondents in sampled classes by tutors through WeChat, a popular mobile app in China. We contacted the full-time tutors of our sampled colleges and classes to send out the link of electronic questionnaire in their WeChat class group. Each IP address was allowed one questionnaire response. Statements of the purpose of the research and assurance of the confidentiality and privacy of participating individuals were placed on the first page of the survey questionnaire. After reading this statement, participants could only complete the questionnaire by clicking "AGREE" to confirm their consent. All participants were told that they had the right to stop the survey at any time. Respondents could get lucky money after finishing online survey in each wave.

As for the comment that “from a Western perspective, that the respondents may have been under some implicit pressure to complete the surveys and/or to provide socially acceptable responses”, we are sorry for the misleading sentence. We have revised this sentence: “Full-time tutors who majored in psychology, education, and administration were trained online at the beginning of the survey, including the aim of the study, the data collection procedures, and privacy protection policy during the survey, et al.”in Lines 284-287. We revised the text in Lines 223, 234-236.

  • While the authors have used null hypothesis significance testing (NHST) approaches the paper is essentially a descriptive one. The aim is to document the change in the key variables during a very specific time in history and there are no formal hypotheses being tested here (are there?).  The p-values associated with the various tests are being used as descriptive rather than inferential statistics andit might be better to simply present the data with confidence intervals and omit the NHST testing. If the tests are to be retained then some form of Type 1 error protection (e.g. Bonferroni-Holm correction) would be advisable.  It would also be appropriate to report indicators of effect sizes since many of the effects in the graphs in Figure 1 appear to be quite modest. 

Response: Many thanks for the comment and suggestion. As requested and suggested by the reviewer, we have made modification as follows:

First, analysis of variance (ANOVA) was performed to test if social mentality varied with time ( See Table 3 in the revised version we have submitted). The null hypotheses of the tests were that the seven social mentalities did not change with time (wave). The p-values of the hypothesis tests were then adjusted by Bonferroni-Holm correction as suggested (α= 0.007). The change of balanced social mentality (BSM) over group variables were tested by repeated measures analysis of variance (RM-ANOVA), with one group variable and time (wave) tested in one model. The null hypotheses of the RM-ANOVA were: (1) the BSM did not change over time; (2) the mean values of BSM did not differ over groups. Also, we adjusted the P values by Bonferroni-Holm correction (α= 0.004) as suggested (See Table 5 in the revised version) for Type 1 error protection. As suggested by the reviewer, we made modification in Lines 276-289.

Second, post hoc analysis was used in order to compare difference of social mentality among five waves and explore specific changing trajectory from wave-1 to wave-5. (See Table 4 in Lines 326-329). The same method was used to display difference within subgroups including university, major, grade, parenting style and harmonious degree of parents. (See Supplementary appendix S1). Furthermore, we added interpretation of results above in Results section in Lines 359-368.

Third, it’s noted that we made multivariate analysis by linear mixed models to explore associates of social mentality in supplementary files (Supplement Appendix S2) we submitted in the first edition. Considering the aim of this study as changing trajectories and difference among waves and subgroups, we have depicted results from multivariate analysis in the Discussion section and made comment on these results in Lines 807-810.

  • The material in lines 318 -328 seems to be a political statement in support of the Chinese government and seems unrelated to the data presented. Similarly on lines 331 and onwards it is not clear what the Chinese policy of placing vaccines on the international market has to do with the promotion of individual mental resilience in students. Lines 391-399 contain more political polemic.  Whether the actions of the Chinese authorities worked well or not will be a matter for historians to decide, the present data say nothing about this.

Response: Many thanks for the comment. We have checked it carefully and agree with the reviewer. As suggested by the reviewer, we have deleted content in Lines 633-644 and Lines 826-834.

  • On line 388 it says “Negative mentality (anxiety, depression, pressure, worry, information-burnout, etc.) was constantly relieved, and positive mentality (peacefulness, serenity, gratitude, happiness, etc.) was constantly reinforced. The optimal mixed mode in this study (see Supplementary Table S2).”But this is not what the data shows as there is first a decline followed by a mild recovery. The graphs in Figure 1 show relatively minor changes across time.

Response: Many thanks for the comment. We agree with the reviewer’s comment here. It’s not suitable to say “Negative mentality (anxiety, depression, pressure, worry, information-burnout, etc.) was constantly relieved, and positive mentality (peacefulness, serenity, gratitude, happiness, etc.) was constantly reinforced.” 

We are very sorry to make a few mistakes in the analysis on the difference among waves 1 to 5. Updated results indicate that not all the indicators of social mentality were significantly different among waves. So we corrected the results for NIM, PPbM, PSM in Table 3 and made modification in Lines 403-405. However, the result for BSM, the indicator of overall social mentality in this study was invariant.

Although the data shows that there is a decline followed by a mild recovery, results of analysis of variance (ANOVA) indicated that the average PIM, NPbM, NSM, BSM values changed significantly from wave1-wave5 (p < 0.001). BSM, the dependent variable in statistical analysis, as a result of the balance between positive social mentality (sum of individual positive mentality and public positive mentality) and negative social mentality (sum of individual negative mentality and public negative mentality) was considered as the indicator of overall social mentality. Paired t-tests in post hoc analysis were used to examine the difference of BSM between waves in detail (See Table 4 in Lines 326-329). Compared to wave-1, the social mentality of wave-2 and wave-3 were decreased significantly (p < 0.001). Compared to wave-2, the social mentality of wave-3 were decreased significantly (p = 0.001). Compared to wave-3, the social mentality of wave-4 were increased significantly (p < 0.001). However, the increase from wave-1 to wave-5 was not significant. 

Therefore, we made modification regarding the results above. First, we have deleted Lines 779-780 and 822-871. Second, considering the aim and length of the manuscript, we made comment on future research direction regarding the associates in Discussion section.

  • Line 406 discusses parenting styles but this was not explained in the method or results sections.

Response: Many thanks for the comment. We are so sorry for not giving results in the context. As requested by the reviewer, we have supplemented results of family variables including family style, parenting style and degree of parents’ relationship in Result section (See Table 2 and Table 5) and in Supplementary files (Appendix S1). We made modification in Lines in 455-464 and 472-481 in Results section. More words are added in Lines 730-750 in Discussion section regarding the results above.

Language and style in the revised version have been polished by a native speaker in English.

Round 2

Reviewer 2 Report

thank you for preparing the revised manuscript 

Author Response

Dear Editor/Reviewers,

We would like to express our great appreciation to you for your valuable comment and support. 

Reviewer 3 Report

I thank the authors for engaging with my review in a positive way.  I understand and accept their conceptualisation of social mentality but I am still at a loss to see how this conceptual position fits with the measurement of social mentality by using traditional measures taken from individuals who, as is evident in the authors’ own data, vary and thus do not reflect some common state of society or a group.  You are still aggregating individual level measurements so I struggle to see how this fundamentally differs from the standard Western approach. We still don’t have a translation of the BDSMQ items so claims about the dimensions being tapped and exactly how they are tapped will remain unclear to the non-Chinese speaking readers.  Having said this, the Western approach to social science is not privileged in anyway so if the authors wish to continue with their position this is, of course, fine.  I do suggest that the BDSMQ scale is presented (or parts of it) to help the reader. 

For future reference, the acronym BDSM has an unfortunate subcultural meaning in the West and the authors might think about revising it to ‘B-DSM’ or similar.  Check out the BDSM Wikipedia entry.

Author Response

Dear Editor/Reviewers,

We would like to express our appreciation to the reviewer for the valuable comments. We have carefully considered and discussed your comments point by point, and made careful corrections and modifications to the article, all of which have been highlighted. Our specific responses are as follows. Its noted that Lines... here refers to the revised manuscript we submitted using the tracking pattern in WPS.

List of action:

  1. “BDSMQ”is revised for “B-DSMQ”in Lines 21, 317, 219, 326, 344 in the text.
  2. “See Supplementary Appendix S1”is added in Line 326.
  3. “1”is revised to “2”in Line 491.
  4. “2”is revised to “3” in Line 822.
  5. “Supplement”is revised to “Suppplementary” in Line 491.
  6. Supplementary Appendix S1 (Translation of some items in B-DSMQ) is added in supplementary files.

Responses to Reviewer 3

  1. I thank the authors for engaging with my review in a positive way.  I understand and accept their conceptualisation of social mentality but I am still at a loss to see how this conceptual position fits with the measurement of social mentality by using traditional measures taken from individuals who, as is evident in the authors’ own data, vary and thus do not reflect some common state of society or a group.  You are still aggregating individual level measurements so I struggle to see how this fundamentally differs from the standard Western approach. We still don’t have a translation of the BDSMQ items so claims about the dimensions being tapped and exactly how they are tapped will remain unclear to the non-Chinese speaking readers.  Having said this, the Western approach to social science is not privileged in anyway so if the authors wish to continue with their position this is, of course, fine.  I do suggest that the BDSMQ scale is presented (or parts of it) to help the reader. 

Response: Many thanks for your comment and suggestion. As you requested and suggested, we have added a translation of some items in B-DSMQ scale in supplementary files (See S1).

  1. For future reference, the acronym BDSM has an unfortunate subcultural meaning in the West and the authors might think about revising it to ‘B-DSM’ or similar.  Check out the BDSM Wikipedia entry.

Response: Many thanks for noting this. We have revised it to ‘B-DSMQ’ and made modifications in Lines 21, 317, 219, 326, 344 in the text. All is highlighted in the revised manuscript.